# Could Self-Control and Emotion Influence Physical Ability and Functional Recovery after Stroke?

**DOI:** 10.3390/medicina57101042

**Published:** 2021-09-30

**Authors:** Yu-Won Choe, Myoung-Kwon Kim

**Affiliations:** 1Department of Rehabilitation Sciences, Graduate School, Daegu University, Jillyang, Gyeongsan 712-714, Korea; choiyuwon@naver.com; 2Department of Physical Therapy, College of Rehabilitation Sciences, Daegu University, Jillyang, Gyeongsan 712-714, Korea

**Keywords:** self-control, emotion, stroke, recovery

## Abstract

*Background and Objectives*: This study was conducted to determine whether self-control and emotions could influence patients’ physical ability and functional recovery after stroke. *Materials and Methods*: Twenty-four patients within eight weeks after a stroke were included in this study (age: 54.04 ± 10.31; days after stroke: 42.66 ± 8.84). The subjects participated in tests at the baseline, four weeks later, and eight weeks later. Subjects were asked to complete the following: (1) self-control level test, (2) positive and negative emotion test, (3) knee muscle strength testing, (4) static balance test, (5) gait measurement, and (6) activities of daily living evaluation. *Results*: The muscle strength of the knee, static balance, gait ability, and the Functional Independence Measure score increased significantly in the stroke patients over time. A significant correlation was noted between the emotion and physical variables in stroke patients. The self-control level was significantly associated with the change in the physical variables in stroke patients over time. *Conclusions*: The self-control level was positively related to the increases in functional recovery of stroke patients with time, while the emotions were related more to the physical abilities.

## 1. Introduction

The incidence of stroke is increasing with the increasing average life span and aging. Despite the advances in acute management, stroke remains a major cause of disability worldwide [1]. Many stroke survivors report long-term disability and reduced quality of life [2,3]. The effects of stroke may include sensory, motor, and cognitive impairment as well as a reduced ability to perform self-care and participate in social and community activities [4,5].

Motor impairment after a stroke typically affects the control of arm and leg movement on one side of the body [6,7]. Stroke patients have disabilities that result from paralysis, and most complain of difficulties in walking [8]. A common feature of walking after a stroke includes decreased gait velocity [9]. Stroke patients also have an increased body sway because of their decreased balance function and distorted standing posture [10,11]. Balance problems are also believed to be common after stroke, and they have been implicated in the poor recovery of the activities of daily living (ADL) and mobility and the increased risk of falls [12]. In addition, complex motor activities have been investigated such as reaching while standing and showed how stroke subjects had more endpoint trajectory variability than healthy [13]. Much of the focus of stroke rehabilitation, in particular the work of physical therapists, is focused on the recovery of physical independence and functional ability during ADL; the ultimate goal of therapy is to improve the function of walking and recovery of balance and movement [7]. Researchers have attempted to achieve the goal by finding the most effective rehabilitation for recovery in stroke patients [1,14]. Various novel stroke rehabilitation techniques for motor recovery have been developed based on basic science and clinical studies [1]. A recent study about neurophysiological mechanisms for enhancing stroke recovery have shown that pharmacological interventions might promote neural plasticity and could potentially enhance the effectiveness of post-stroke motor therapies. In addition, therapies that directly stimulate the PNS or CNS may enhance neuroplasticity during post-stroke rehabilitation, and these might help patients with stroke overcome their motor impairments [15]. In the post-acute stage (<3 months after stroke), the focus shifts to neurorehabilitation that may include exercise combined with technology such as robot-assisted training [16] or neuromuscular electrical stimulation for UL rehabilitation [17], which has been shown to play a key role in functional recovery. Clinical trials aiming at enhancing training-based neuroplasticity have incorporated different principles of motor learning [18] and treatment interventions [19]. On the other hand, the effectiveness of interventions among patients with stroke varies widely because of the heterogeneous mechanisms underlying motor recovery [1]. Currently, no single standard intervention has been identified that is effective for recovering function after stroke [20].

Previous studies suggested that stroke rehabilitation programs should include meaningful, repetitive, intensive, and task-specific movement training in an enriched environment to promote neural plasticity and functional recovery [18,21]. In addition, most recovery is believed to be made in the first stages after stroke [5,22]. Hence, the physical and psychological condition of the patient in the acute stage is essential. On the other hand, even if the early period of stroke rehabilitation is critical to their recovery, physical therapists cannot enforce meaningful, repetitive, intensive, and task-specific training on the patients. Therefore, the patients themselves have to actively participate in the rehabilitation for their recovery after stroke, even if the training is difficult. Active participation in rehabilitation could be related to self-control. Self-control is conceptualized as the extent of a person’s self-perception of having control over events and ongoing situations and reflects the perception of an ability to manage them [23,24]. Furthermore, feeling control is important for psychological adjustment, which has been the strongest predictor of a person’s ability to carry out behaviors for achieving the desired goal by taking action [23,25]. Several studies investigated individual differences in self-control, which are known as self-regulation in psychology, and reported that high levels of self-control are linked to positive outcomes [23,26]. In addition, self-control, a psychological factor, should not be overlooked in the rehabilitation of stroke survivors [23].

Self-control is currently one of the most popular topics of psychological research. Its popularity is driven by the fundamental role of behavioral self-regulation in maintaining appropriate functioning in day-to-day life [27]. The definition of self-control is usually straightforward and is typically associated with regulating emotions, desires, and behavior [28]. Nęcka et al. introduced five behavioral components of self-control: (1) goal maintenance, or the ability to consider one’s intentions and long-term plans; (2) proactive control, defined as the ability to develop plans, prioritize goals, analyze consequences, and predict obstacles; (3) initiative and persistence, understood as the ability to set goals and take actions to follow one’s decisions without unnecessary delays; (4) switching and flexibility, or the ability to adjust to changing circumstances and manage attention during the performance of appropriate actions; (5) inhibition and adjournment, defined as the ability to inhibit emotional reactions and refrain from immediate, impulsive behavior [27]. These behavioral components of self-control could be important when stroke patients participate in rehabilitation. Tangney et al. reported that self-control is involved in managing stress, focusing attention on specific tasks, modifying their responses according to their needs, and planning and performing the activities required to achieve long-term goals [29]. Numerous research findings showed that high self-control is related to better outcomes in various areas such as physical health, substance dependence, higher scholastic performance, coping better with frustration and stress, personal finances, and criminal offending outcomes [30,31,32]. In addition, a previous study suggested using the self-control level to predict people’s achievement [33]. Physical goals, such as increasing muscle strength, balance, gait ability, and independence of ADL are also achievements that stroke patients want to accomplish. Self-control is a complex phenomenon that involves many psychological functions and has a strong impact on everyday performance [27]. In South Korea, stroke patients who are hospitalized receive treatment at least five days a week. The long-term rehabilitation is a difficult task, particularly in patients with neurological problems, such as stroke. Self-control, which includes goal maintenance, proactive control, initiative and persistence, switching and flexibility, and inhibition and adjournment, will be necessary for rehabilitation and recovery outcomes.

Positive emotion is also recognized as an essential factor in people’s achievement [33]. Positive emotion is measured by the positive and negative experiences in daily life [34]. Salovey et al. reported that positive emotions and healthy outcomes might be linked through multiple pathways [35]. Positive emotional states may promote healthy perceptions, beliefs, and physical well-being. Recently, many studies focused on the physical function and emotional problems in stroke patients [36,37,38,39]. This is because psychological factors greatly influence the physical and functional activities [40]. Previous studies suggested that positive emotions significantly affect functional capacity [41,42]. A higher positive emotion was significantly associated with higher motor, cognition, and ADL ability in stroke patients and with the gains in functional status post-stroke [43]. A previous study organized numerous research data (cross-sectional, longitudinal, and experimental studies) to identify the effects of positive emotion [44]. In the review article, numerous studies showed that people who experience frequent positive emotions achieved more in diverse domains than those who did not. The evidence suggests that positive emotion may cause many desirable characteristics, resources, and successes [44]. Positive people appear to be relatively more likely to seek approach goals. The interaction of cognitive judgments about the desire for a change with the propensity for a positive emotion is a ripe area of inquiry for the future [44]. These people can have goals and experience positive emotion due to the progress they make toward those goals. Positive people also perform well in many areas that require motivation and persistence. At the same time, specific dissatisfaction can also motivate happy people to work for a change and pursue new directions [44]. Several theories have proposed that high positive emotion is related to increased recovery following an acute medical event [45,46,47]. Other studies found that negative emotions decreased muscle strength and could adversely affect the physical function [36,40,48,49]. In addition, many studies reported that negative emotions, such as depression, anxiety, emotionalism, and post-traumatic stress disorder interfere with recovery and cause stroke patients to lack the motivation to participate in rehabilitation, leading to decreased participation in ADL [20,50,51].

Many researchers have examined the important factors related to physical ability and functional recovery in stroke patients. Recently, the psychological effects have been debated widely in stroke rehabilitation. Previous studies did not consider the stroke patient’s self-control and positive and negative emotion as important factors of physical ability and functional recovery. Moreover, most stroke rehabilitation still focuses on physical problems without addressing the psychological effects. This study examined whether self-control and emotions could influence physical ability and functional recovery after a stroke. We expect that there is a positive correlation between self-control level and function recovery in stroke patients.

## 2. Materials and Methods

### 2.1. Participants

Thirty-three stroke patients participated in this study. Nine patients dropped out. The data of 24 patients were analyzed. All participants were hospitalized in each of the three hospitals. The hospitals are located in Daegu and Pohang, South Korea, and all hospitals have stroke rehabilitation unit. Before commencing the study, all subjects understood the content of the study and signed an informed consent form. This study complied with the ethical standards of the declaration of Helsinki. The ethical committee of Daegu University approved this study (1040621-202101-HR-023).

The subjects were required to meet the following inclusion criteria: (1) patients with hemiplegia following a first unilateral stroke (ischemic and hemorrhagic) [52]; (2) acute stroke patients within three months after the stroke; (3) patients had a mini-mental state examination-Korean (MMSE-K) score of 24 or greater, indicating no cognitive impairment [8]; (4) patients could understand and follow the therapist’s directions [53]; (5) patients could participate sufficiently in conversation [36]; (6) the ability to stand and walk 10 m. People were excluded from the study if they had other neurological conditions in addition to stroke, had unstable cardiovascular disease, or had other serious diseases that precluded participation in the study [54].

All patients participated in the rehabilitation programs in each hospital. They engaged in at least two hours of physical therapy (muscle strengthening, endurance training, balance exercise, mat activities, and gait training), one hour of occupational therapy (functional movement re-education, ADL training, and swallowing therapy), and one hour of functional electrical stimulation (FES) therapy on each weekday and at least one hour of physical therapy, 30 min of occupational therapy, and 30 min of FES therapy on the weekend. Speech-language therapy was provided as needed.

### 2.2. Experimental Procedure

Statistically, this is a repeated-measures design study. Epidemiologically, this is single cohort longitudinal study. All measurements were evaluated at the baseline, four weeks later (first follow-up), and eight weeks later (second follow-up). After the subjects agreed to participate in this experiment, they underwent a self-control level test, positive and negative emotion test, and physical ability tests (muscle strength, static balance, gait, and ADL). The order of tests was as follows: (1) self-control level test, (2) positive and negative emotion test, (3) muscle strength, (4) static balance, (5) gait, and (6) ADL. All tests were performed in a quiet environment with an examiner. The participants had a three-minute break between tests. Figure 1 presents a flowchart of the study.

### 2.3. Measurements

#### 2.3.1. Self-Control

This study used the Korean version of the Brief Self-Control Scale (BSCS) to identify the self-control level in stroke patients. Tangney et al. developed the BSCS [29]. Studies reported that BSCS has high validity and high reliability in self-control [55,56,57]. The internal consistency and test–retest reliability coefficients of the scale were 0.85 and 0.87, respectively [58]. Hong et al. translated the BSCS into the Korean version and examined the internal consistency (Cronbach’s α = 0.78); they reported that the Korean version of the BSCS had high convergent validity [59]. The Korean version of the BSCS is an 11-item, single-factor scale based on self-reporting. Individuals rate each item from 1 (not at all) to 5 (very much) points using a Likert-type scale. The score was reversely calculated to determine the negatively worded items (reversed items). The maximum and minimum score of BSCS was 55 and 11, respectively.

#### 2.3.2. Positive and Negative Emotion

This study used the Korean version of the Positive and Negative Affect Schedule (PANAS) to identify the positive and negative emotions in stroke patients. The PANAS examined positive experiences and negative experiences in daily life. Watson et al. developed the PANAS, and the scale is highly internally consistent and stable at appropriate levels over a two-month period [34]. Lee et al. examined the reliability (internal consistency 0.84) and validity of the Korean version of the PANAS [60]. The Korean version of the PANAS consisted of 20 items (positive emotion, 10 items and negative emotion, 10 items), and the scale is based on self-reporting. Individuals rate each item from 1 (not at all) to 5 (extremely) points using a Likert-type scale. The score range was between 50 (maximum) and 10 (minimum) in each emotion.

#### 2.3.3. Muscle Strength

The muscle strength of the lower extremity was assessed using a handheld dynamometer (PowerTrack II MMT, COMMANDER, JTECH Medical, Salt Lake City, UT, USA). This device measures the force generated by a group of muscles involved. The intra-rater reliabilities were excellent for the knee extension strength, and the inter-rater reliabilities were excellent for the knee extension [61].

The subject was instructed to perform maximum isometric contraction during the dynamometer measurements [62]. The dynamometer was placed perpendicular to the tested limb [62]. For knee extensors testing, the subject was in the sitting position with their knee and hip flexed to 90°. The examiner stabilized the subject’s thigh to avoid movement of the tested limb. The handheld dynamometer was placed just proximal to the ankle on the anterior surface of the leg [63]. A one-minute rest was given between two consecutive trials to prevent muscle fatigue. The highest force produced during each session was recorded [62].

#### 2.3.4. Static Balance

A force platform (FDM SX, Zebris Medical GmbH, Isny, Germany) was used to measure the static balance. Previous studies reported a good ICC (>0.90) for this system [64,65]. This force platform was a 55 cm × 40 cm plate with a 40 cm × 30 cm sensor surface with 1920 pressure sensors. The sampling rate for the device was 120 Hz. The force platform system measured the center of pressure (COP) path length and COP sway velocity. A higher COP path length and sway velocity indicated inferior static standing balance performance [66]. The weight-bearing proportion of each foot to the whole was also measured.

In this test, all subjects stood on the platform barefoot with their hands next to their bodies and staring at the wall in front of them [65,67]. The location of each foot was recorded to ensure that the feet were positioned at the same place during reassessment testing. During testing with their eyes open, they were asked to stare at a 15 cm diameter dot placed 3 m ahead [67]. The examiner closely supervised the subject during the test to prevent falls. Data were captured for ten seconds, and five successful trials were recorded with a thirty-second rest between trials [65]. The mean value of five successful trials was used for data processing.

#### 2.3.5. Gait Measurement

The GAITRite (CIR Systems, Inc, Franklin, NJ, USA) system was used to evaluate the temporal parameters, such as the gait velocity and cadence, as well as spatial parameters, such as single-limb support [68]. The gait velocity is the distance per time (cm/s), and the cadence is steps per minute. The single-limb support represents a phase in the gait cycle when the bodyweight is supported entirely by one limb while the contralateral limb swings forward. The GAITRite mat exhibits excellent reliability for most of the temporospatial gait parameters measured in older subjects [69].

For the measurements, the subjects were instructed to stand 3 m away from the electronic carpet and walk across the carpet at a comfortable walking speed, stopping after walking 3 m past the electronic carpet. An examiner closely supervised the subject during the test to prevent falling. The measurements were repeated three times with a one-minute break between the measurements to minimize the potential for bias caused by muscle fatigue. The average value of the three trials was calculated and recorded [68].

#### 2.3.6. Activities of Daily Living

The Functional Independence Measure (FIM) was used to evaluate the independence of the activities of daily living (ADL). The FIM is an 18-item measurement tool that explores an individual’s physical, psychological, and social function. The tool is used to assess a patient’s level of disability and the changes in patient status in response to rehabilitation or medical intervention [70,71]. The FIM contains 18 items composed of 13 motor tasks and five cognitive tasks. The tasks were rated on a seven-point ordinal scale ranging from total assistance (complete dependence) to total independence (complete independence). The score ranged from 18 (lowest) to 126 (highest), indicating the level of function. The tasks that were evaluated using the FIM included bowel and bladder control, transfer, locomotion, communication, social cognition, and the following six self-care activities (feeding, grooming, bathing, upper body dressing, lower body dressing, and toileting). Hsueh et al. reported that the FIM has excellent internal consistency (FIM motor subscale) (Cronbach’s alpha = 0.88) and excellent concurrent validity (excellent correlation between the FIM motor subscale and the 10-item version of the Barthel Index; r = 0.92) in acute stroke [72].

### 2.4. Date Analysis

The sample size for this study was calculated using the G* Power program 3.1.0 (G power program Version 3.1, Heinrich-Heine University Dusseldorf, Dusseldorf, Germany) [73,74]. Based on the data from the pilot study, the estimated sample size was obtained from a power at 0.08 and significant level at 5%. This resulted in a sample size of 18 subjects being required. Thirty-three participants were recruited to account for a potential dropout.

SPSS 20.0 software (SPSS Inc., Chicago, IL, USA) was used for all statistical analyses. A Kolmogorov–Smirnov test was used to determine the type of distribution for all variables. One-way repeated ANOVA was applied to determine the significant difference in the self-control score, positive and negative emotions score, and physical variables over time. The Bonferroni correction method was used to control for multiple comparisons. The Pearson’s correlation coefficient was used for correlation analysis between variables. The correlation was calculated between self-control, positive and negative emotions, muscle strength, static balance, gait, ADL, and changes in variables. The level of significance was 0.05.

## 3. Results

The general characteristics of the subjects are shown in the Table 1.

Changes in physical variables in stroke patients during eight weeks are shown in Table 2 and Figure 2. The velocity, cadence, and affected side single support increased significantly for eight weeks (*p* < 0.05). The COP path length and COP sway velocity decreased significantly for eight weeks. Weight bearing on the affected side increased significantly (*p* < 0.05). The knee extensor strength increased significantly for eight weeks (*p* < 0.05). The FIM score increased significantly for eight weeks (*p* < 0.05). There was no significant change in self-control, positive emotion, and negative emotion for eight weeks (*p* > 0.05) (Table 3) (Figure 3).

No significant correlation was observed between the self-control and physical variables at the baseline, four weeks, and eight weeks (*p* > 0.05) (Table 4).

No significant correlation was observed between the positive emotion and velocity, cadence, affected side single support, COP path length, weight-bearing on the affected side, knee extensor strength, and Functional Independence Measure at the baseline, four weeks, and eight weeks (*p* > 0.05). A significant correlation was observed between positive emotion and COP sway velocity at the eight weeks (*p* < 0.05) (Table 5).

Significant correlation was observed between the negative emotion and velocity, cadence, affected side single support, COP path length, COP sway velocity, knee extensor strength, and Functional Independence Measure (*p* < 0.05) (Table 6).

Significant correlation was observed between the self-control at the baseline and changes in velocity, cadence, COP sway velocity, weight bearing on the affected side, and knee extension muscle strength during four weeks (*p* < 0.05). Significant correlation was observed between the self-control at the baseline and changes in velocity, cadence, and COP sway velocity during eight weeks (*p* < 0.05) (Figure 4). No significant correlation was observed between the positive emotion at the baseline and changes in physical variables during four and eight weeks (*p* > 0.05) (Figure 4). No significant correlation was observed between the negative emotion at the baseline and changes in physical variables during four and eight weeks (*p* > 0.05) (Figure 4).

## 4. Discussion

This study is the first to determine that self-control and emotion could influence the physical ability and functional recovery in acute stroke patients. The results showed that the self-control level affects the functional recovery and improvement after stroke, while the emotions are related more to the physical abilities.

Physical abilities, including muscle strength of the lower extremities, static balance, gait, and independence of ADL, were improved significantly in acute stroke patients over eight weeks. The stroke patients who participated in this study were in the early period of rehabilitation (weeks after stroke: three weeks (*n* = 2); four weeks (*n* = 2); five weeks (*n* = 6); six weeks (*n* = 7); seven weeks (*n* = 6); and eight weeks (*n* = 1)). Many studies reported that most stroke patients show considerable recovery of function over the first few months [75,76,77,78]. Recovery was reported to be fastest in the first few weeks after a stroke but suggests that it can continue beyond the first three months [79]. Numerous studies showed significant increases in the muscle strength of the upper and lower extremities within the first few weeks or months after stroke [80,81,82,83]. Many studies reported that static balance, dynamic balance, and gait function were increased significantly within a few weeks or months after stroke [79,84,85]. In addition, previous findings showed that the gain of ADL in stroke patients who participated in intensive rehabilitation was most significant during the first few weeks or months after a stroke [86,87]. Given all these findings, the muscle strength, static balance, gait function, and the independence of ADL appear to be improving in the first few weeks or months after stroke.

There were no significant changes in self-control, positive emotion, and negative emotion score in acute stroke patients during eight weeks. Hence, the self-control, positive emotion, and negative emotion levels in stroke patients were consistent over time. This was attributed to the emotional experience. No critical events, such as falls, serious changes in health condition, cognitive problem, or death of a family member, which could change the self-control, positive emotion, and negative emotion levels, occurred during the study. On the other hand, these results are inconsistent with those of a previous study. Seale et al. examined the changes in positive emotion over a three-month follow-up [47]. They found that 35.6%, 29.2%, and 35.2% of participants reported an increase, a decrease, and no change in positive emotion, respectively [47]. Other studies also reported that positive emotion is a dynamic process that can vary with time [43,88]. This study attributed these inconsistent results to the difference in environment. The previous study measured the emotion in stroke patients at discharge and the three-month follow-up. In contrast, the present study measured the emotions in stroke patients who were hospitalized. These differences in the environment may explain the inconsistent results between studies.

No significant correlation was observed between self-control and physical abilities (muscle strength of knee, static balance, gait, and ADL) at the baseline, four weeks, and eight weeks. Hence, high self-control level does not lead to better physical abilities in stroke patients. Previous studies reported that the physical abilities are determined by the stroke severity [89,90,91], not the self-control level. From all these considerations, it was assumed that self-control is not related to the physical abilities of acute stroke patients.

In this study, a negative correlation was observed between the positive emotion score and COP sway velocity. On the other hand, there was no significant correlation between positive emotion score and knee muscle strength, gait variables, and the FIM score. These results mean that positive emotion is related to the static balance. The results of this study are consistent with previous studies. Previous studies suggested that positive emotions significantly affect the functional capacity measured by testing lifting, postural tolerance, and repetitive movement [41,42]. On the other hand, the mechanism of the correlation between positive emotion score and physical ability remains unclear. LaLumiere et al. reported that emotional stimulation influences the synaptic plasticity of the brain and descending tracts from the brain [92]. This could affect the motor system. Given these findings, it was assumed that positive emotion is positively related to the static balance of acute stroke patients.

A negative correlation was observed between the negative emotion score and the knee strength. A positive correlation was noted between the negative emotion score and static balance variables (COP path length and COP sway velocity). A negative emotion was negatively associated with the gait variables (velocity, cadence, and affected side single support). A negative correlation was noted between the negative emotion score and the FIM score. These results mean that negative emotions are related to the knee muscle strength, static balance, gait, and independence of the ADL. These are consistent with previous studies. Studies found that negative emotions decreased the muscle strength and could adversely affect the physical function [36,40,48,49]. Other studies reported a negative correlation between depression and functional ability and motor skills in stroke patients [93,94]. From these considerations, it was assumed that the negative emotion is negatively related to the physical abilities of acute stroke patients.

The reason why negative emotion was more correlated with physical abilities than positive emotion was because of differences in measurement items. This study used the PANAS scale to examine the patient’s positive and negative emotions. The negative emotions in PANAS were composed of the following: upset, distressed, nervous, jittery, guilty, ashamed, hostile, irritable, scared, and afraid. The stroke patients experienced these feeling easily during hospitalization after stroke. The positive emotions in PANAS were composed of the following: interested, alert, attentive, excited, enthusiastic, inspired, proud, determined, strong, and active. The stroke patients might have difficulty feeling these emotions, such as inspired, excited, or enthusiastic, and these emotions are related to feelings that are not easy to experience during hospitalization regardless of the patient’s physical ability.

In this study, there was a significant positive correlation between the self-control score and the increases in knee muscle strength. A significant positive correlation was noted between the self-control score and the static balance improvement (COP sway velocity and weight bearing on the affected side). A significant positive correlation was noted between the self-control score and the increases of the gait variables (velocity and cadence). These results mean that the self-control level is positively related to the improvement of the knee muscle strength, static balance, and gait. Choi et al. examined the relative contributions of self-control and positive emotion on achievement. Across five studies (*n* = 1130), they reported that self-control and positive emotions are important predictors of achievement [33]. In addition, self-control is more strongly related to achievement than positive emotion. This finding holds for college students, middle-school students, East Asian adults, and North American adults, suggesting that it is cross-cultural and robust across age and measurement [33]. Improvement of physical ability and functional recovery are important goals for stroke patients. Goal achievement, such as increasing muscle strength, balance, gait function, and independence of the ADL could be affected by the self-control level. A previous study reported that self-control is involved in managing stress, focusing attention on specific tasks, modifying the responses according to the needs, and planning and performing activities required to achieve long-term goals [29]. Stroke patients could be motivated by their long-term goals. Many rehabilitation professionals commonly believe that patients’ motivation plays an important role in determining the outcome [95,96,97,98]. The motivation to achieve a goal, such as better walking and independence of ADL, could promote the physical recovery of stroke patients. For these reasons, stroke patients who have a higher self-control level tend to achieve their goal more than patients with a lower self-control level. In addition, many research findings showed that high self-control is related to better achievement in various areas such as physical health, substance dependence, higher scholastic performance, coping better with frustration and stress, personal finances, and criminal offending outcomes [30,31,32]. Kim and Park reported that self-control has a significant, positive emotion on the ADL in stroke patients [23]. This previous finding is consistent with the present study. From these considerations, the self-control level appears to be increasing the functional recovery after stroke. Given these findings, it was assumed that the self-control level is positively related to functional recovery in acute stroke patients over time. On the other hand, there was no correlation between the self-control level and the increases in FIM score. The FIM includes upper and lower extremity motor function items, and the upper limb function has a significant impact on the change in score on this test. However, in this study, three patients could not contract their upper extremity muscles in any of the tests, and there were no changes in the muscle strength over time. The three patients had relatively high self-control levels in the baseline (24, 26, and 36), four weeks (25, 25, and 37), and eight weeks (26, 27, and 37). It was assumed that this study was conducted with a small sample size, and the data from the patients could affect these results.

No significant correlation was observed between the positive emotion at the baseline and changes in physical variables during four and eight weeks. No significant correlation was observed between the negative emotion at the baseline and changes in physical variables during four and eight weeks. These results mean that positive and negative emotions are not related to the functional recovery. On the other hand, these results are inconsistent with those of a previous study. Ostir et al. examined the effects of positive emotions on the functional ability of stroke patients [43]. Eight hundred and twenty-three stroke patients participated in the study at discharge and the three-month follow-up. A higher discharge positive emotion (high CES-D score) was significantly associated with higher motor and cognition FIM ratings and a higher total FIM rating after three months [43]. These results indicate that positive emotions are associated with gains in functional status post-stroke. In people with stroke, increases in positive emotions over a three-month period were significantly associated with an increased likelihood of recovery of functional status [47]. Kim and Park suggested that stroke patients who experience less depressed moods and stress have more self-control and maintain a higher ADL level [23]. This study attributed these inconsistent results to the difference in measurement scale and environmental factors. Given the results, further discussion is needed to identity whether emotions affect the functional recovery in stroke patients.

In summary, this study found that the emotions were related to the physical abilities, while the self-control level was not. The self-control level is positively related to increased functional recovery in stroke patients over time. High self-control helps improve the knee muscle strength, static balance, and gait function after stroke.

This study had several limitations. First, the sample size was small, and there was no control group. Second, this study did not control the effects of the drug, rehabilitation quality, and environment. Third, this study did not include testing for depression and did not report physical conditions of the patients after stroke. Lastly, this study could not identify the long-term effects of self-control and emotions after eight weeks. Given these limitations, additional studies of the long-term effects, widening the subject range, and a well-controlled study will be needed.

## 5. Conclusions

This study examined whether self-control and emotions could influence the physical ability and functional recovery of stroke patients. The self-control level is positively related to the increases in functional recovery in acute stroke patients over time, while the emotions are related to the physical ability. Given these findings, the self-control level in acute stroke is essential for functional recovery, and less negative emotions promote physical ability. These findings highlight the need to include measures of self-control and emotions to improve functional recovery after a stroke.

## Figures and Tables

**Figure 1 medicina-57-01042-f001:**
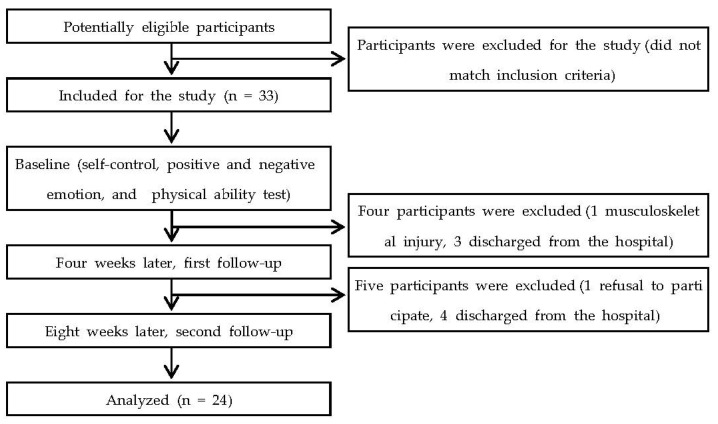
Study flow chart.

**Figure 2 medicina-57-01042-f002:**
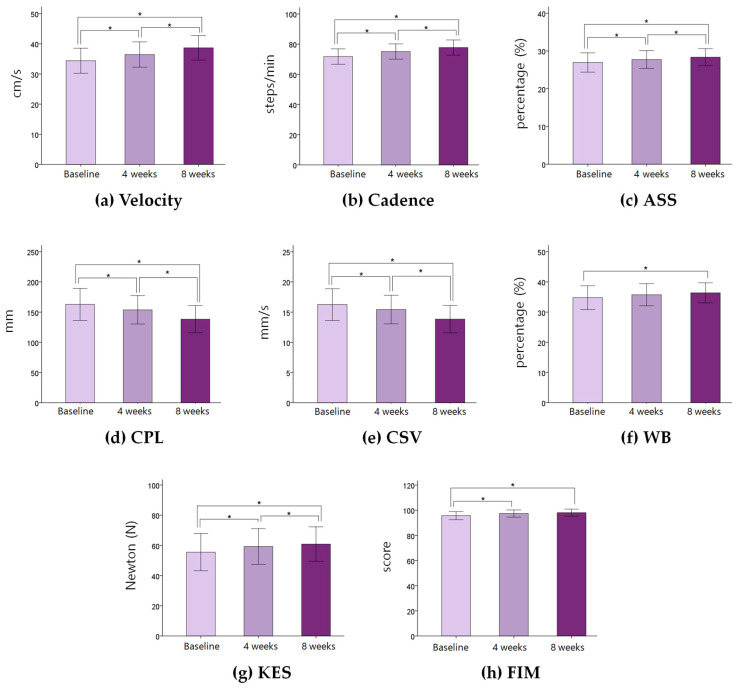
Changes in physical variables between baseline, 4 weeks, and 8 weeks. Each bar represents the mean and standard error of the mean. Asterisk (*) means statistical significance. ASS: affected side single support, CPL: center of pressure path length, CVS: center of pressure sway velocity, WB: weight bearing on the affected side, KES: knee extensor strength, FIM: Functional Independence Measure.

**Figure 3 medicina-57-01042-f003:**
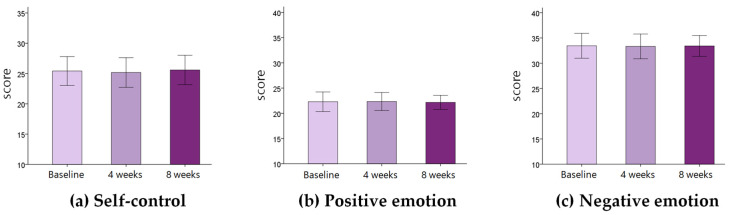
Changes in psychological variables between baseline, 4 weeks, and 8 weeks. Each bar represents the mean and standard error of the mean.

**Figure 4 medicina-57-01042-f004:**
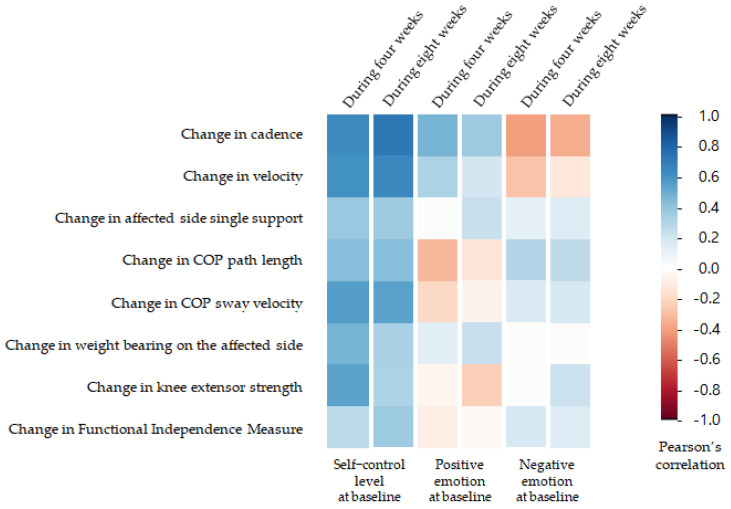
Correlation matrix of self-control level, positive emotion, negative emotion, and changes in physical variables.

**Table 1 medicina-57-01042-t001:** General characteristics of the subjects.

Variable	
Age (year)	54.04 (10.31) ^1^
Duration (days after stroke)	42.66 (8.84)
Height (cm)	164.88 (7.90)
Weight (kg)	65.21 (9.42)
MMSE-K ^2^ (score)	25.91 (1.31)
Positive emotion (score)	22.2 (4.7)
Negative emotion (score)	33.4 (6.0)
Sex (male/female)	14/10
Type (hemorrhage/infarction)	13/11
Paretic side (right/left)	11/13

^1^ Mean (± standard deviation). ^2^ Mini Mental State Examination-Korean.

**Table 2 medicina-57-01042-t002:** Changes in physical variables during eight weeks.

	Baseline	Four Weeks	Eight Weeks	F	*p*
Velocity (cm/s)	34.3 (10.1) ^1^	36.4 (10.2)	38.6 (9.9)	89.5	0.00 *
Cadence (steps/min)	71.7 (12.4)	75.0 (12.3)	77.7 (12.3)	115.9	0.00 *
ASS (%) ^2^	26.9 (6.2)	27.7 (5.7)	28.3 (5.5)	22.6	0.00 *
CPL (mm) ^3^	162.8 (64.8)	153.7 (57.6)	138.2 (55.0)	21.7	0.00 *
CSV (mm/s) ^4^	16.2 (6.4)	15.4 (5.8)	13.8 (5.5)	22.3	0.00 *
WB (%) ^5^	34.7 (9.6)	35.7 (8.9)	36.3 (8.1)	6.7	0.01 *
KES (N) ^6^	55.5 (30.0)	59.2 (29.0)	60.8 (28.0)	36.1	0.00 *
FIM (score) ^7^	95.6 (7.8)	97.3 (7.0)	98.0 (6.8)	23.4	0.00 *

^1^ Mean (±standard deviation). ^2^ Affected side single support. ^3^ Center of pressure path length. ^4^ Center of pressure sway velocity. ^5^ Weight bearing on the affected side. ^6^ Knee extensor strength. ^7^ Functional Independence Measure. * Statistical significance *p* < 0.05.

**Table 3 medicina-57-01042-t003:** Changes in psychological variables during eight weeks.

	Baseline	Four Weeks	Eight Weeks	F	*p*
Self-control (score)	25.4 (5.8) ^1^	25.1 (5.9)	25.5 (5.9)	1.1	0.32
Positive emotion (score)	22.2 (4.7)	22.3 (4.3)	22.1 (3.3)	0.1	0.83
Negative emotion (score)	33.4 (6.0)	33.3 (6.0)	33.4 (5.0)	0.0	0.93

^1^ Mean (± standard deviation).

**Table 4 medicina-57-01042-t004:** Correlation between self-control and physical variables.

	Baseline Self-Control	Four Weeks Self-Control	Eight Weeks Self-Control
Velocity (cm/s)	0.19 ^1^	0.26	0.29
Cadence (steps/min)	0.08	0.16	0.22
ASS (%) ^2^	0.24	0.29	0.30
CPL (mm) ^3^	−0.20	−0.32	−0.38
CSV (mm/s) ^4^	−0.18	−0.32	−0.39
WB (%) ^5^	0.18	0.26	0.22
KES (N) ^6^	0.18	0.24	0.22
FIM (score) ^7^	0.19	0.28	0.31

^1^ Correlation coefficient. ^2^ Affected side single support. ^3^ Center of pressure path length. ^4^ Center of pressure sway velocity. ^5^ Weight-bearing on the affected side. ^6^ Knee extensor strength. ^7^ Functional Independence Measure.

**Table 5 medicina-57-01042-t005:** Correlation between positive emotion and physical variables.

	Baseline Positive Emotion	Four Weeks Positive Emotion	Eight Weeks Positive Emotion
Velocity (cm/s)	0.28 ^1^	0.35	0.31
Cadence (steps/min)	0.23	0.35	0.29
ASS (%) ^2^	0.33	0.38	0.29
CPL (mm) ^3^	−0.37	−0.36	−0.42
CSV (mm/s) ^4^	−0.36	−0.37	−0.43 *
WB (%) ^5^	0.22	0.23	0.24
KES (N) ^6^	0.26	0.31	0.23
FIM (score) ^7^	0.27	0.34	0.31

^1^ Correlation coefficient. ^2^ Affected side single support. ^3^ Center of pressure path length. ^4^ Center of pressure sway velocity. ^5^ Weight bearing on the affected side. ^6^ Knee extensor strength. ^7^ Functional Independence Measure. * Statistical significance *p* < 0.05.

**Table 6 medicina-57-01042-t006:** Correlation between negative emotion and physical variables.

	Baseline Positive Emotion	Four Weeks Positive Emotion	Eight Weeks Positive Emotion
Velocity (cm/s)	−0.44 ^1^ *	−0.45 *	−0.37
Cadence (steps/min)	−0.39	−0.44 *	−0.39
ASS (%) ^2^	−0.43 *	−0.43 *	−0.33
CPL (mm) ^3^	0.47 *	0.45 *	0.43 *
CSV (mm/s) ^4^	0.45 *	0.45 *	0.44 *
WB (%) ^5^	−0.32	−0.32	−0.27
KES (N) ^6^	−0.45 *	−0.44 *	−0.30
FIM (score) ^7^	−0.42*	−0.43*	−0.35

^1^ Correlation coefficient. ^2^ Affected side single support. ^3^ Center of pressure path length. ^4^ Center of pressure sway velocity. ^5^ Weight bearing on the affected side. ^6^ Knee extensor strength. ^7^ Functional Independence Measure. * Statistical significance *p* < 0.05.

## Data Availability

The data presented in this study are available on reasonable request from the corresponding author.

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
