# Peer review of "Could Self-Control and Emotion Influence Physical Ability and Functional Recovery after Stroke?"

_medicina, 2021, doi:10.3390/medicina57101042_

Round 1

Reviewer 1 Report

Dear Authors, 

Thank you for the opportunity to revise your manuscript. I enjoyed reading your manuscript.  I thank the authors for their effort in producing this research during the COVID-19 pandemic. 

Please find below constructive comments that may help to improve the clarity of the manuscript and the scientific soundness.  

# Introduction: 
Line 32-40: I agree that stroke affect motor abilities. Recently, more complex motor activities have been investigated such as reaching while standing, showing how stroke subjects had more endpoint trajectory variability than healthy. Authors may add Tomita Y, et al., Stability of reaching during standing in stroke. J Neurophysiol. 2020 May 1;123(5):1756-1765. doi: 10.1152/jn.00729.2019

Line 46. Authors may add a recent work about the need of understanding neurophysiological mechanism for enhancing stroke recovery: Piscitelli D. Neurorehabilitation: bridging neurophysiology and clinical practice. Neurol Sci. 2019 Oct;40(10):2209-2211. doi: 10.1007/s10072-019-03969-2.

Line 82. Specify the areas that are improved through high self-control.

Line 88. Please add the reference or specify the country. For example, in high-income countries the health systems differ from low-middle countries.

Line 127. I would use the word “addressing” instead of “understanding.”

Line 128. Please state the objectives and the hypothesis. What the Authors expect from their project? 

# Materials and Methods 

Line 132-135 Move the section about the sample size under the paragraph “2.5. Date analysis ”. Explain how the Effect size of 0.1 was estimated.

Line 136: Which Hospital? Were stroke-units? 

Line 149: Authors should explain and detailed the type of physical and occupational therapy administrated to the patients.  

Line 156: Statistically this is a repeated-measures design study. Epidemiologically, this is single cohort longitudinal study. 

Line 198. An average of three measurement for the muscle strength for each muscle group would be more a reliable assessment. Why was not performed?

# Results

Table 1. Use “sex” instead of “gender”

In Table 2 all the variables show significant differences with one-way ANOVA. Authors should present the pairwise comparison, Bonferroni corrected.  Please carefully control the statistics e.g., for Velocity, the standard deviation seems to overlap. Add all the degrees of freedom.

# Discussion 

Lines 344 – 347. The sentence is unclear. 
What does it mean that emotions were more related to the physical abilities “at that moment”?  The results show no difference of positive emotion, and negative emotion for eight weeks (p>0.05) and the only a significant correlation was observed between positive emotion and COP sway velocity at the eight weeks. In table 5 the Correlation between negative emotion and physical variables are all less than 0.5 underling a Moderate association.

Lines 407 – 408. The statement is unclear. What does it mean “at that moment” ?

Lines 475 – 476. Again, the statement is unclear. What does it mean “at that moment” ?

# Abstract 

Add in the Methods the age (mean and SD) and the time from stroke of the participants

The conclusion about the “the emotion related more to the physical abilities at that moment.” Is unclear. Please modify it. 

Author Response

Dear Reviewer,

Thank you very much for reviewing our paper. Thanks to your review, we could improve the clarity of the manuscript and the scientific soundness. Also, we checked all your suggestions and tried to revise it as best as we can. It would be much appreciated if you could go over it.

# Introduction: 
Line 32-40: I agree that stroke affect motor abilities. Recently, more complex motor activities have been investigated such as reaching while standing, showing how stroke subjects had more endpoint trajectory variability than healthy. Authors may add Tomita Y, et al., Stability of reaching during standing in stroke. J Neurophysiol. 2020 May 1;123(5):1756-1765. doi: 10.1152/jn.00729.2019

= Thanks for your suggestion. We added the reference and sentence.

Line 46. Authors may add a recent work about the need of understanding neurophysiological mechanism for enhancing stroke recovery: Piscitelli D. Neurorehabilitation: bridging neurophysiology and clinical practice. Neurol Sci. 2019 Oct;40(10):2209-2211. doi: 10.1007/s10072-019-03969-2.

= Thanks for your suggestion. We added a recent review article about neurophysiological mechanisms for enhancing stroke recovery.

Line 82. Specify the areas that are improved through high self-control.

= According to your suggestion, we specify the areas that are improved through high self-control.

Line 88. Please add the reference or specify the country. For example, in high-income countries the health systems differ from low-middle countries.

= Thanks for your suggestion. We added the country to specify the country.

Line 127. I would use the word “addressing” instead of “understanding.”

= According to your suggestion, we revised the word.

Line 128. Please state the objectives and the hypothesis. What the Authors expect from their project? 

= Thanks for your suggestion. We added what we had expected from our project.

# Materials and Methods 

Line 132-135 Move the section about the sample size under the paragraph “2.5. Date analysis ”. Explain how the Effect size of 0.1 was estimated.

= According to your suggestion, we moved the section and added sentence about how to estimate the sample size using G*power program.

Line 136: Which Hospital? Were stroke-units? 

= We added information about hospitals’ location and stroke units.

Line 149: Authors should explain and detailed the type of physical and occupational therapy administrated to the patients.  

= Thanks for your review. We detailed the type of physical and occupation therapy administrated to the patients.

Line 156: Statistically this is a repeated-measures design study. Epidemiologically, this is single cohort longitudinal study. 

= Thanks for letting us know. According to your suggestion, we added sentences in experimental procedure.

Line 198. An average of three measurement for the muscle strength for each muscle group would be more a reliable assessment. Why was not performed?

= Thanks for letting us know. We followed a previous study’s method that used the same device as we used to examine the muscle power of knee extensor group.

# Results

Table 1. Use “sex” instead of “gender”

= According to your suggestion, we revised the word.

In Table 2 all the variables show significant differences with one-way ANOVA. Authors should present the pairwise comparison, Bonferroni corrected.  Please carefully control the statistics e.g., for Velocity, the standard deviation seems to overlap. Add all the degrees of freedom.

= Thanks for letting us know. To present the pairwise comparison and Bonferroni corrected, we added graphs (Fig. 2, 3). The graphs show the differences between three different time points (baseline, four weeks, and eight weeks).

# Discussion 

Lines 344 – 347. The sentence is unclear. 
What does it mean that emotions were more related to the physical abilities “at that moment”? The results show no difference of positive emotion, and negative emotion for eight weeks (p>0.05) and the only a significant correlation was observed between positive emotion and COP sway velocity at the eight weeks. In table 5 the Correlation between negative emotion and physical variables are all less than 0.5 underling a Moderate association.

= Thanks for your review. The reason why we used “at that moment” in the article is to emphasize “the current/present state” But, according to your suggestion, we revised the sentence in order to clarify the meaning. Also, we revised the sentences in results, discussion, conclusion, and abstract.

= Thanks for letting us know. The Correlation between the emotion and physical variable was low, however it was statistically significant, so that we wanted to show summarized results of this study in the first paragraph of discussion. There was a statistically significant correlation between negative emotion and physical variable in baseline and 4 weeks. In 8 weeks, there was a statistically significant correlation between negative emotion and balance variables. There was a statistically significant correlation between positive emotion and only one balance variable in 8weeks. To clarity the results for readers, we show summarized results first and then explained more details in each part of discussion that shows correlation between negative emotion, positive emotion, and physical variables.

Lines 407 – 408. The statement is unclear. What does it mean “at that moment” ?

= Thanks for your review. We revised the sentence

Lines 475 – 476. Again, the statement is unclear. What does it mean “at that moment” ?

= Thanks for your review. We revised the sentence.

# Abstract 

Add in the Methods the age (mean and SD) and the time from stroke of the participants.

= According to your suggestion, we added information about subjects’ age (54.04±10.31) and days after stroke (42.66±8.84) in Abstract.

The conclusion about the “the emotion related more to the physical abilities at that moment.” Is unclear. Please modify it. 

= Thanks for your review. We revised the sentence.

* We attached PDF in order to prove our English editing

Reviewer 2 Report

In this interesting study , a significant correlation was noted between the positive and negative emotion and physical variables in stroke patients. The self-control level was significantly associated with the change in the physical variables in stroke patients over time. The relation between mood disorders and stroke recovery is a critical question , this work suggest that therapy targeting this condition may improve stroke recovery . This is consistent with the last therapeutic attempt using serotonin re uptake inhibitor like prozac or paroxetine . Although the sample size cannot assess this distinction further study are needed to clarify the relationship between mood disorders and lesion location , right versus left cortical versus subcortical . 

Author Response

Dear Reviewer,

In this interesting study , a significant correlation was noted between the positive and negative emotion and physical variables in stroke patients. The self-control level was significantly associated with the change in the physical variables in stroke patients over time. The relation between mood disorders and stroke recovery is a critical question , this work suggest that therapy targeting this condition may improve stroke recovery . This is consistent with the last therapeutic attempt using serotonin re uptake inhibitor like prozac or paroxetine . Although the sample size cannot assess this distinction further study are needed to clarify the relationship between mood disorders and lesion location , right versus left cortical versus subcortical . 

= Thank you very much for reviewing our paper. Your suggestion paved the way to developing our further research methods. According to your suggestion, we will make effort to clarify the relationship between mood disorders and sites of lesion in our further study.

* We attached PDF in order to prove our English editing.

Reviewer 3 Report

This study tried to invetsigate the effect of positive and negative emotions on physical ability and functional recovery after stroke.

This study is explorative in nature and that needs to be emphasised in the manuscript. The results and the conclusions are based mainy on the use of Pearson's correlation coefficient.

In addition, there is no information on how depressed the patients were after stroke. Depression afetr stroke is very common in stroke patients, and this needs to be reported and taken into account in the analyisis. Otherwise, there is no content in the use of "positive" and "negative" emotions.

What was the pre-stroke physical condition of the patients? I'd argue that the resuts on the obcetive physical meassurment depend on the pre-stroke physical strenght of the patients.

On the Methods section the authors state: "Bonferroni method was used for post-hoc comparisons". Boferroni is a method to correct for multiple comparisons, not a method for post-hoc analysis. By the way, in exploratory stduies you are allowed not to use correction methods for multiple comparisons.

Author Response

Dear Reviewer,

Thank you very much for reviewing our paper. We checked all what you stated and tried to revise it as best as we can. It would be much appreciated if you could go over it.

This study tried to invetsigate the effect of positive and negative emotions on physical ability and functional recovery after stroke.

This study is explorative in nature and that needs to be emphasised in the manuscript. The results and the conclusions are based mainy on the use of Pearson's correlation coefficient.

In addition, there is no information on how depressed the patients were after stroke. Depression afetr stroke is very common in stroke patients, and this needs to be reported and taken into account in the analyisis. Otherwise, there is no content in the use of "positive" and "negative" emotions.

= Thanks for letting us know. We didn’t evaluate the patients’ depression level after stroke. However, we evaluated patients’ negative emotion using the PANAS scale. The negative emotion items of the PANAS scale include upset, distressed, nervous, jittery, guilty, ashamed, hostile, irritable, scared, and afraid, which are related to depression. We added patients’ positive and negative emotion score in the Table 1.

What was the pre-stroke physical condition of the patients? I'd argue that the resuts on the obcetive physical meassurment depend on the pre-stroke physical strenght of the patients.

= Thanks for the review and suggestion. We didn’t evaluate the patients’ pre-stroke physical condition. But, we will make sure to include that in our later study.

On the Methods section the authors state: "Bonferroni method was used for post-hoc comparisons". Boferroni is a method to correct for multiple comparisons, not a method for post-hoc analysis. By the way, in exploratory stduies you are allowed not to use correction methods for multiple comparisons.

= Thanks for letting us know. To present the pairwise comparison and Bonferroni corrected, we added graphs (Fig. 2, 3). The graphs show the differences between three different time points (baseline, four weeks, and eight weeks).

* We attached PDF in order to prove our English editing

Round 2

Reviewer 1 Report

Dear Authors.,

Thank you for the careful revision of your manuscript. I feel it is much better now.

I have only a minor revision about Figure 2 and Figure 3. Where it is appropriate add the Standard deviation (SD) or the Standard error of the mean (SEM) on the bar plots.See for example:  Cumming, G., Fidler, F. & Vaux, D.L. J. Cell. Biol. 177, 7–11 (2007).

Congratulation with your work.

Author Response

Dear Reviewer,

Thank you very much for reviewing our paper.

Thanks to your review, we could improve our research. We checked your suggestion and tried to revise it as best as we can.

I have only a minor revision about Figure 2 and Figure 3. Where it is appropriate add the Standard deviation (SD) or the Standard error of the mean (SEM) on the bar plots.See for example:  Cumming, G., Fidler, F. & Vaux, D.L. J. Cell. Biol. 177, 7–11 (2007).

We added the standard error of the mean (SEM) on the bar of the figures and explained it. It would be much appreciated if you could go over it.

Thank you very much.

Sincerely yours,

Reviewer 3 Report

I still think that there are important limitations of this study - namely not testing for depression and not reporting the physical condition of the patients after stroke. These need to be mentioned in the Discussion in a special paragraph (Limitations...). 

In the Methods section, please correct the sentence

"Bonferroni’s method was used for post-hoc analysis" -> "Bonferroni correction method was used to control for multiple comparisons". 

Are Figure 2 and 3 representing data from a single subject or from a group of subjects. This needs to be mentioned in the figure caption. If the Figures summarise group data, you need to add bars representing appropriate meassure of devaition, such as standard deviation, standard error of the mean etc. 

Author Response

Dear Reviewer,

Thank you very much for reviewing our paper.

Thanks to your review, we could improve our research. We checked all your suggestions and tried to revise it as best as we can.

I still think that there are important limitations of this study - namely not testing for depression and not reporting the physical condition of the patients after stroke. These need to be mentioned in the Discussion in a special paragraph (Limitations...). 

Thanks for your suggestion. We added the sentence (limitation of our study) on the last paragraph in the discussion.

In the Methods section, please correct the sentence

"Bonferroni’s method was used for post-hoc analysis" -> "Bonferroni correction method was used to control for multiple comparisons". 

Thanks for letting us know. According to your suggestion, we revised the sentence.

Are Figure 2 and 3 representing data from a single subject or from a group of subjects. This needs to be mentioned in the figure caption. If the Figures summarise group data, you need to add bars representing appropriate meassure of devaition, such as standard deviation, standard error of the mean etc. 
Thanks for your suggestion. We added the standard error of the mean (SEM) on the bar of the figures and explained it.

It would be much appreciated if you could go over it.

Thank you very much.

Sincerely yours,